# Comparison of patient exit interviews with unannounced standardised patients for assessing HIV service delivery in Zambia: a study nested within a cluster randomised trial

Kombatende Sikombe [1,2] Jake M Pry [1] Aaloke Mody [3] Brian Rice,[2] Chama Bukankala,[1] Ingrid Eshun-Wilson [3] Jacob Mutale,[1] Sandra Simbeza,[1] Laura K Beres,[4] Njekwa Mukamba,[5] Mpande Mukumbwa-Mwenechanya,[1] Daniel Mwamba,[1] Anjali Sharma,[5] Alison Wringe,[6] James Hargreaves,[2] Carolyn Bolton-Moore,[1,7] Charles Holmes,[8] Izukanji T Sikazwe,[1] Elvin Geng[3]

For numbered affiliations see end of article.

Correspondence to
Kombatende Sikombe;
kombatende.sikombe@cidrz.org

## ABSTRACT

**Objectives** To compare unannounced standardised patient approach (eg, mystery clients) with typical exit interviews for assessing patient experiences in HIV care (eg, unfriendly providers, long waiting times). We hypothesise standardised patients would report more negative experiences than typical exit interviews affected by social desirability bias.

**Setting** Cross-sectional surveys in 16 government-operated HIV primary care clinics in Lusaka, Zambia providing antiretroviral therapy (ART).

**Participants** 3526 participants aged ≥18 years receiving ART participated in the exit surveys between August 2019 and November 2021.

**Intervention** Systematic sample (every nth file) of patients in clinic waiting area willing to be trained received pre-visit training and post-visit interviews. Providers were unaware of trained patients.

**Outcome measures** We compared patient experience among patients who received brief training prior to their care visit (explaining each patient experience construct in the exit survey, being anonymous, without manipulating behaviour) with those who did not undergo training on the survey prior to their visit.

**Results** Among 3526 participants who participated in exit surveys, 2415 were untrained (56% female, median age 40 (IQR: 32–47)) and 1111 were trained (50% female, median age 37 (IQR: 31–45)). Compared with untrained, trained patients were more likely to report a negative care experience overall (adjusted prevalence ratio (aPR) for aggregate sum score: 1.64 (95% CI: 1.39 to 1.94)), with a greater proportion reporting feeling unwelcome by providers (aPR: 1.71 (95% CI: 1.20 to 2.44)) and witnessing providers behaving rude (aPR: 2.28 (95% CI: 1.63 to 3.19)).

**Conclusion** Trained patients were more likely to identify suboptimal care. They may have understood the items solicited better or felt empowered to be more critical. We trained existing patients, unlike studies that use 'standardised patients' drawn from outside the patient population. This low-cost strategy could improve patient-centred service delivery elsewhere.

## STRENGTHS AND LIMITATIONS OF THIS STUDY

⇒ This study used standardised patients (SPs) to assess chronic care in which actual, rather than simulated, patients were trained before their upcoming clinic visits.
⇒ Traditional SP techniques require a trained simulated patient to visit multiple clinics, a strategy more appropriate for episodic care.
⇒ Modified SP approaches can address the challenge of integrating patient experience into routine public health, a crucial quality indicator for governments and funders.
⇒ We trained patients to assess care quality (eg, waiting times, rude providers), and compared their responses with traditional untrained exit surveys in 16 facilities in Zambia.
⇒ Training remains challenging as we did not include participants who were illiterate, had poor recall ability or potentially struggled with comprehension.

**Trial registration number** Assessment was nested within a parent study; www.pactr.org registered the parent study (PACTR202101847907585).

## BACKGROUND

Because of improved testing, linkage and treatment to meet the global 95-95-95 treatment targets (95% of HIV-positive patients know their status, 95% are on treatment and 95% have suppressed viral loads),[1] retention in care has become a major obstacle to improving HIV treatment outcomes, and health systems in low-income settings like Zambia have sought to shift their public health response by designing and delivering high-quality and patient-centred HIV care.[2–7] Efforts to improve service quality and

patient experience require systematic measurement of the patient experience to guide facility responses as poor patient experience has been shown to lead to disengagement from care.[8–12] Health policymakers and donors, such as the President's Emergency Plan for AIDS Relief, have invested in clinical metrics to assess care quality in Zambia and the wider region, but to a lesser extent in non-clinical metrics like patient experience.[13] These metrics can be critical for guiding efforts to improve retention in care by ensuring an informed response to improving quality of care and patient centredness.

Accurate and pragmatic measurement of the patient experience poses a range of challenges. Patient experience exit surveys are prone to social desirability bias because of power dynamics in healthcare. Empirical studies of satisfaction, for example, are widely believed to overestimate patient satisfaction.[14] This may be particularly true where provider–patient relationships are traditional and hierarchical. Delaying surveys for some time after the encounter is theorised to ameliorate social desirability bias, but in turn may exacerbate bias due to simple inability to remember—thus creating recall bias.[7 15] Other methods such as direct clinical observations of care pose practical difficulties.[14 16] For example, direct observations may be intrusive and therefore may not reflect everyday functionality of a health facility. Care provided under direct observations may be of higher quality as behaviour may be influenced by observation, a phenomenon often known as the 'Hawthorne effect'.[14 16]

Standardised patients (SPs), also known as 'mystery clients' or 'simulated patients' have largely been used to assess quality of care in developed countries, as well as in assessing customer service in the retail industry.[17] SPs can be resource-intensive and require training, but reduce potential for recall bias, social desirability bias and Hawthorne effects, providing an opportunity for optimal assessment of patient satisfaction among people receiving HIV care.[7 15 18] They have largely been used for episodic care where a highly skilled and well-trained person poses as a client by making one visit to multiple facilities. This approach holds promise for assessing the patient experience in HIV care but poses pragmatic challenges when assessing the quality of chronic care in which a patient makes multiple visits and may compromise efficiency at, already overburdened, facilities.[19–24] In this study, we report on the development and evaluation of a modified SP approach in which we trained real patients (trained exit clients (TECs)) to report on certain characteristics of encounters, and rate key components of care such as waiting times, communication, respectfulness of providers and privacy.

## METHODS
### Study design and setting
This study seeks to compare two different methods for assessing patient experience: standard exit survey and those reported by patients who had brief training on the items before the clinical encounter and to whom the clinic was blinded. The assessment was nested within a parent study: the Leveraging Person-Centred Public Health (PCPH) to improve HIV outcomes in Zambia Study (www.pactr.org; PACTR202101847907585), a stepped-wedge cluster randomised trial that occurred between August 2019 and November 2021. The aim of the overall PCPH Study was to assess the impact of introducing healthcare workers (HCWs) to a patient-centred care (PCC) curriculum and mentoring them on PCC principles to improve retention and viral suppression in HIV care. In this nested substudy, we compared cross-sectional surveys of patient experience using two different survey methods: adapted standardised approach (TECs) versus traditional exit surveys.

### Population
The substudy reported here included 16 health facilities in Lusaka, Zambia, operated by the Ministry of Health (MOH) and receiving technical assistance from the Centre for Infectious Diseases Research in Zambia (CIDRZ)—a Zambian non-governmental organisation as well as a part of the larger parent study. We surveyed adults aged 18 years and over who were accessing antiretroviral therapy (ART) at study facilities. Exit survey patients were selected in a systematic sample (every nth file varied by facility size) at the time of exit from the clinic. Trained patients were recruited in the waiting room for their visit, underwent brief training and then answered survey questions on exit from their encounter. Participants attending an HIV care visit on the day, able to recall events and comprehend study participant recruitment details (as assessed using the comprehension assessment tool) and able to read and write (assessed using literacy tool) were eligible for inclusion.

### Procedures and measurements
#### Survey instrument
For both survey methods, we developed a patient experience instrument based on a previously validated tool developed and used in Kenya: The Wachira Physician-Patient Communication Behaviours Scale.[25–27] This survey assessed elements of patient experience including how they were greeted, communicated to and overall experience. We included additional questions to capture, for example, patient reports of witnessing rude behaviour, receiving appropriate medications and availability of laboratory results. Prior to use in this study, we performed cognitive interviews among 20 participants to assess consistency in understanding questions in English, Bemba and Nyanja. Surveys were forward and back translated to ensure consistency across the three languages. The survey tools for trained and untrained clients were identical. Research assistants were trained by the first author in recruitment, training and administering of the TEC and untrained exit client (UEC) survey in all 16 facilities. The provincial and district health management teams were informed about the unannounced TEC

survey as well as the UEC survey. The study team sensitised all facility staff at the start of the study, but HCWs were not aware of who specific TECs were.

## Procedures for TECs and UECs

Efforts to 'standardise' assessment of the quality and nature of care in HIV differ from most previously standardised patient or mystery client work in that HIV care is longitudinal as opposed to episodic or acute care. Under these circumstances, the more conventional standardised patient where a single trained actor can present to multiple different care facilities as a simulated patient with a defined set of symptoms or complaints to assess a single episode of care is not feasible. For example, a patient would have to either register as a new patient or have a false 'file' introduced into the paper and electronic medical records—which was deemed infeasible and undesirable.

Instead of simulated patients, we recruited existing patients already receiving care at a particular facility and then subsequently trained them on the concepts of quality of care according to the MOH manual on Quality Improvement for HCWs in Zambia. To avoid disclosing their trained status, patients were recruited on the day of their visit prior to them entering the triage area (ie, the first point of contact with HCWs). Those who consented underwent single one-on-one training session for 40–60 min where they were sensitised to the study instrument (which was the same for both TECs and UECs), the MOH care standards and strategies on being natural yet observant during their clinic visit for that day according to the standard SP approach. These procedures were meant to ensure patients had a clear and uniform understanding on what they should expect during a high-quality patient visit and were attentive to these critical aspects relative to these standards. Immediately after this training, the TEC presented themselves to their facility and completed their visit as they normally would. After their clinic encounter, participants then completed the exit survey in a private area.

For the UEC surveys, we took a systematic (every n$^{th}$, varied by facility size) sample among the patients leaving the facility after attending the clinic on the survey day. Patients were approached by study staff after the visit using a recruitment script to determine their eligibility and were administered the survey after granting consent in a private area.

For both TEC and UEC, all interviews and surveys were conducted in either English, Bemba or Nyanja depending on the participant's preference. Given the extra time commitments required for the training, TEC participants were given K100 (~$5) for the time spent during training as well as a light snack during the survey administration.

## Statistical analysis

To assess the association between training and response for each question, we conducted unadjusted and adjusted Poisson regression for each question separately.[28] We then assessed the overall association between training

and total sum score. We used descriptive statistics to characterise patient characteristics and report survey responses. In these analyses, most of the survey responses were reverse coded to identify when respondents reported a negative experience. Results for individual questions (binary response) represent prevalence ratios for reporting a lapse in care. To assess the sum score (count data), we used Poisson regression, estimating the rate ratio for reporting lapses in care. All models were adjusted, given potential differences in survey participants related to different recruitment strategies using mixed-effects regression, adjusted for age, sex, education, care status at the time (ie, continuously retained in care vs returning to care after disengagement/lost to follow-up (LTFU)), secular time (using cubic splines), allowing random effects at the facility level. We present these results for the overall population as well as stratified by different predefined patient subgroups. Lastly, we used bubble plots to compare summary assessments of the patient experience at the facility level using TECs versus UECs. All analyses were performed using STATA V.14MP (StataCorp, College Station, Texas, USA). This substudy represents a secondary analysis and no formal power calculations were performed for this outcome.

## Patient and public involvement

Survey questions were developed through a cognitive process with recipients of care. Study implementation guidance was conducted as part of routine CIDRZ partnership with the Zambian MOH through a human-centred design workshop.[29] CIDRZ engages with implementing partners and affected communities in health facilities, including people living with HIV often represented by neighbourhood health representatives. Although patients were not directly involved in the design of the parent study intervention or the analysis presented here, all study activities were guided by a Scientific Advisory Board with representation from the MOH and a representative of recipients of HIV care. Dissemination of study results is ongoing.

## RESULTS
### Characteristics of health facilities and patients

We approached 4375 clients (2955 in the untrained and 1420 in the trained), and 3526 participated, of which 2415 (55.2%) completed experience surveys as UECs (56% female, median age was 40 years (IQR: 32–47 years)) and 1111 (32%) completed experience surveys as TECs (50% female with a median age 37 years (IQR: 31–45 years)). Reasons for non-participation included unavailability at the time due to other commitments. Sixteen per cent of UECs and 40% of TECs had been lost to care and were returning to care on the day of the survey. Education levels differed between UEC and TEC with 47% and 58% reporting completion of secondary level of education, respectively (table 1). UEC and TEC were similar for HIV enrolment WHO stage with the largest proportion

**Table 1** Sociodemographic characteristics of untrained exit and trained exit clients

| Characteristics | Level | Untrained exit clients n=2415 (68%) | Trained exit clients n=1111 (32%) |
|---|---|---|---|
| Sex, n (%) | Female | 1355 (56) | 553 (50) |
| | Male | 1060 (44) | 558 (50) |
| Age, median (IQR) | | 40 (32–47) | 37 (31–45) |
| Age category, n (%) | <30 years | 453 (19) | 258 (23) |
| | 30–40 years | 828 (34) | 416 (37) |
| | 40–50 years | 815 (34) | 304 (27) |
| | >50 years | 319 (13) | 133 (12) |
| Education category | None | 132 (5) | 36 (3) |
| | Primary | 654 (27) | 166 (15) |
| | Secondary | 1134 (47) | 645 (58) |
| | University | 150 (6) | 100 (9) |
| | Missing | 307 (13) | 151 (14) |
| HIV enrolment stage | WHO stage 1 | 1173 (49) | 533 (48) |
| | WHO stage 2 | 314 (13) | 147 (13) |
| | WHO stage 3 | 355 (15) | 162 (15) |
| | WHO stage 4 | 27 (1) | 7 (1) |
| | Missing | 546 (23) | 262 (24) |
| Care status at survey visit | In care | 2038 (84) | 664 (60) |
| | Returning to care | 377 (16) | 447 (40) |
| Marital status | Single | 257 (11) | 167 (15) |
| | Married | 1361 (56) | 575 (52) |
| | Divorced | 248 (10) | 108 (10) |
| | Widowed | 173 (7) | 81 (7) |
| | Unknown | 41 (2) | 20 (2) |
| | Missing | 335 (14) | 160 (14) |
| Facility size | <1000 patients | 591 (25) | 245 (22) |
| | 1000–5000 patients | 897 (37) | 485 (44) |
| | >5000 patients | 927 (38) | 381 (34) |

HIV, Human immunodeficiency virus; IQR, Interquartile range; WHO, World Health Organization.

enrolling at WHO stage 1 and similar in terms of marital status.

Table 2 shows the absolute responses for TEC and UEC. Although most patients reported a good experience, across the questions, between 5% and 25% of patients reported poor experiences in care. For example, when asked if their HIV care provider gave them as much information about their health as they wanted, 13.4% (UEC) vs 24.6% (TEC) of patients reported not being provided with sufficient information about their health. Similarly, between 9.6% vs 18.8% of patients reported that their HIV care provider was not spending the right amount of time with them at their visit, and 6.8% vs 16.4% reported witnessing rude behaviour.

### Effects of training on response patterns: sum score and prevalence ratios

In adjusted models, TECs overall reported poor experiences in care: 1.64 times as frequently as UEC respondents (sum score rate ratio: 1.64 (95% CI: 1.39 to 1.94)) (figure 1 and online supplemental table 1), and reported an increased prevalence of poor experiences in care quality compared with untrained across almost all questions. For example, among TECs compared with UECs, there was an increased prevalence of reports of not being greeted in a way that made them feel welcome (adjusted prevalence ratio (aPR): 1.71 (95% CI: 1.20 to 2.44)), reporting being dissatisfied with all their HIV care providers during their HIV care visit (aPR: 2.06 (95% CI: 1.61 to 2.63)) and witnessing any providers behaving rudely during their visit (aPR: 2.28 (95% CI: 1.63 to 3.19)) (figure 1 and online supplemental table 1).

**Table 2** Survey responses by training status

| Factor | Level | Untrained exit client n (%) | Trained exit client n (%) |
|---|---|---|---|
| Did your HIV care provider greet you in a way that made you feel comfortable? | Yes | 2249 (93.1) | 980 (88.2) |
| | No | 166 (6.9) | 131 (11.8) |
| Did your HIV care provider listen to what you said? | Yes | 2328 (96.4) | 1039 (93.5) |
| | No | 79 (3.3) | 64 (5.8) |
| | Refused | 8 (0.3) | 8 (0.7) |
| Did your HIV care provider give you as much information about your health as you wanted? | Yes | 2092 (86.6) | 838 (75.4) |
| | No | 323 (13.4) | 273 (24.6) |
| Did your HIV care provider allow you to ask questions? | Yes | 2082 (86.2) | 887 (79.8) |
| | No | 326 (13.5) | 222 (20) |
| | Refused | 7 (0.3) | 2 (0.2) |
| Did your HIV care provider spend the right amount of time with you? | Yes | 2179 (90.2) | 900 (81) |
| | No | 232 (9.6) | 209 (18.8) |
| | Refused | 4 (0.2) | 2 (0.2) |
| Overall, how did you feel about the care you received today? | Happy | 2231 (92.4) | 983 (88.5) |
| | Unhappy | 178 (7.4) | 123 (11.1) |
| | Refused | 6 (0.2) | 5 (0.4) |
| Overall, were you satisfied with all your HIV care providers today? | Yes | 2206 (91.4) | 906 (81.5) |
| | No | 208 (8.6) | 202 (18.2) |
| | Refused | 1 (0.0) | 3 (0.3) |
| I witnessed HIV care providers behaving rudely during my visit today. | No | 2251 (93.2) | 928 (83.5) |
| | Yes | 163 (6.8) | 182 (16.4) |
| | Refused | 1 (0.0) | 1 (0.1) |
| Were your lab results lost? | No | 2143 (88.7) | 985 (88.7) |
| | Yes | 268 (11.1) | 126 (11.3) |
| | Not picking up | 4 (0.2) | 0 (0) |
| Were you able to pick up your medicine today? | Yes | 2366 (98.0) | 1087 (97.8) |
| | No | 48 (2.0) | 24 (2.2) |
| | Not picking up meds | 1 (0.0) | 0 (0) |

HIV, Human immunodeficiency virus.

## Impact of training across age, sex and gender to differences in responses

In stratified analysis of the impact of training on the sum score, training was consistently associated with increased identification of poor experiences in care across all subgroups apart from those aged 50 years or older and those with no education. We also observed that training had a larger impact among females compared with males, those with primary education only and among individuals presenting at smaller facilities (figure 2). We observed similarities in responses on the impact of training on different age categories, sex, care status and different levels of education when we looked at individual questions except for the question on providers spending the right amount of time where we found that females were twice as likely to report lapses with care compared with males (online supplemental figure 1). Using TECs gave worse assessments of patient experience at the facility level regardless of facility size compared with UECs (figure 3 and online supplemental figure 2).

## DISCUSSION

Disengaged patients often express a disconnect between their care expectations and the provider's style; hence, experience is bound to vary across facilities.[8] This disconnect can lead to dissatisfaction with HIV services which can often lead to patients dropping out of care.[8 11 30] Brief training for patients living with HIV on how to evaluate the quality and experience of routine care changed patient experience reports compared with untrained patients using the same instrument. Patients who underwent brief training identified more lapses in care across most questions. Women and young people were more likely to report critical responses after training—consistent with the idea that those who feel least empowered

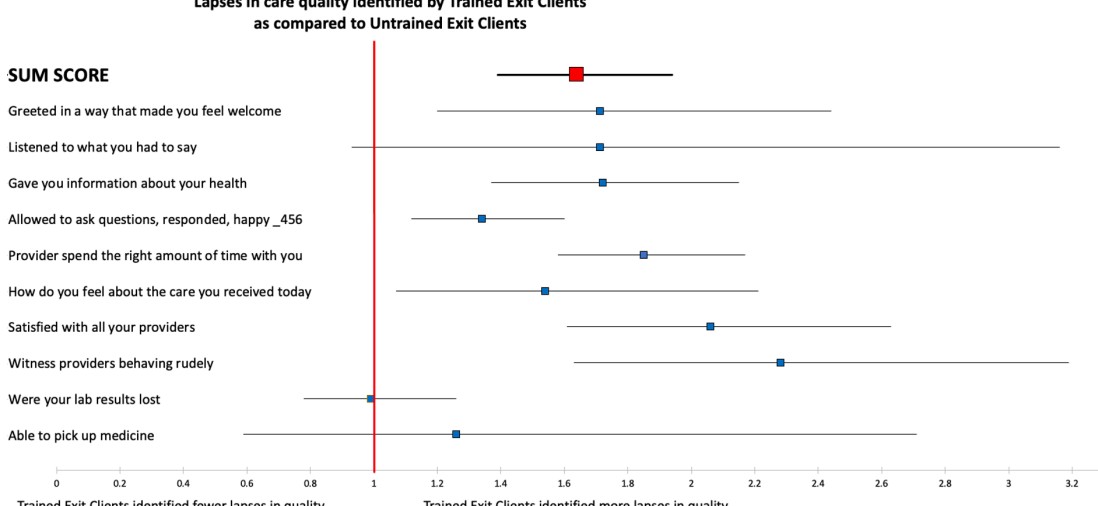

**Figure 1** Forest plot comparing responses from trained exit clients (TECs) relative to untrained exit clients (UECs) on 10 measures of clinic experience. Points indicate the rate ratio (for sum score) or prevalence ratio (for all others) for identifying a lapse in care in TEC surveys as compared with UEC. The sum score represents the total number of binary responses (yes vs no) across all clients in one group shown as a rate ratio. The red line indicates a rate or prevalence ratio of 1 and values greater than this indicate more lapses in care identified in TECs. Results are based on mixed-effects models adjusted for age, sex and education with a random effect at the facility.

underwent the biggest change. Differences were also bigger for questions in which social desirability is likely to operate. For example, larger differences were observed for witnessing rude behaviour, while no differences were observed for more objective questions such as whether laboratory results were lost.

Improving HIV health outcomes requires new strategies that minimise methodological biases and includes everyone the patient encounters during their visit, including clinical officers, doctors, nurses, data clerks

and lay HCWs. Our TEC approach could contribute to getting a true reflection of how much value patients place on things such as effective communication, being greeted appropriately, or being treated with care and respect at all these different touch points. Involving patients in their own care and design of health services has been linked to improved HIV care retention and patient outcomes, such as higher viral suppression rates.[31–33] As progress is being made towards the Joint United Nations Programme on HIV/AIDS 95-95-95 targets, the global HIV sector is

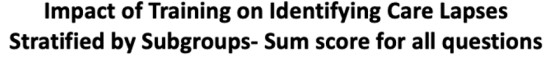

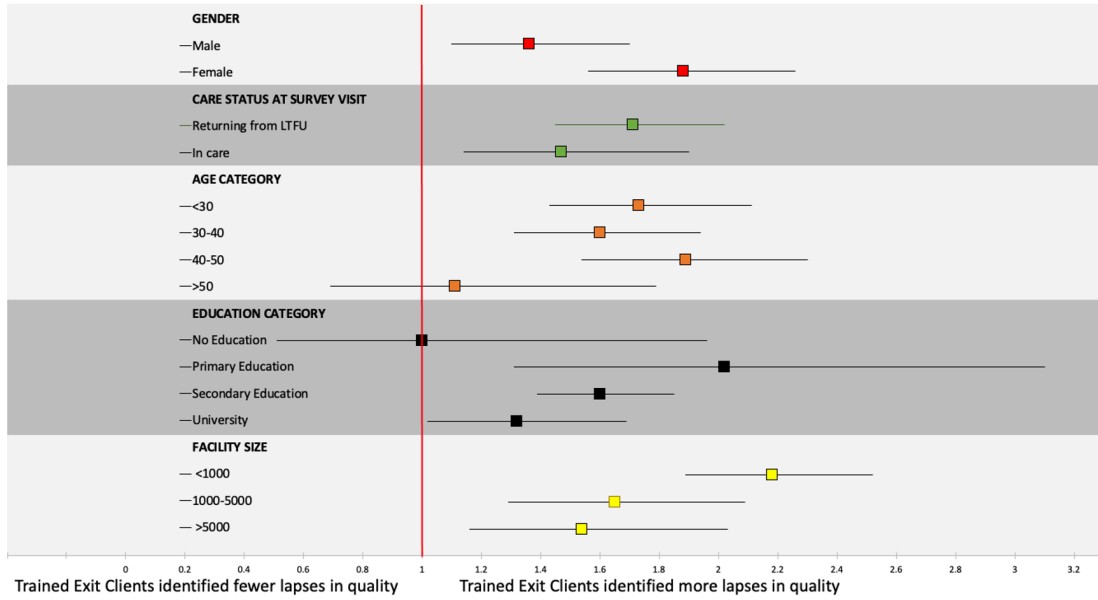

**Figure 2** Impact of training on identifying care lapses stratified by subgroups (N=3480). When all questions were collapsed into a sum score among trained exit clients, females were more likely to report lapses in care quality than males. We observed some level of interaction for care status, age category, education category and facility size. LTFU, lost to follow-up.

**Overall Sum Score – Trained vs. Untrained Exit Survey by Facility**

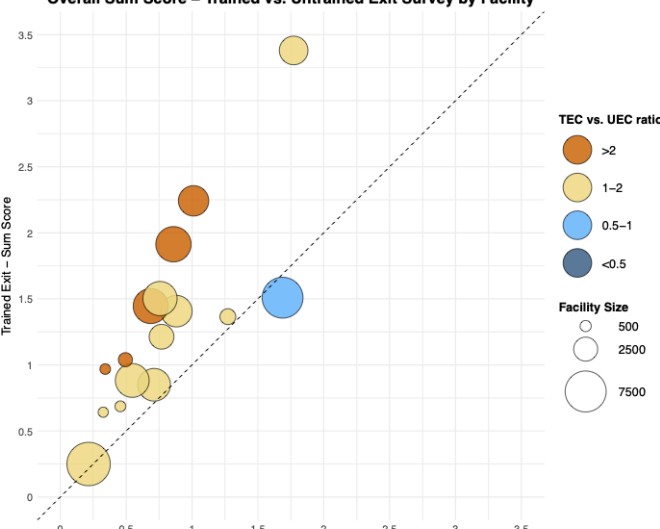

**Figure 3** Bubble plot showing trained exit sum score versus untrained exit sum score. Each bubble represents a single facility's performance. Each bubble's size indicates the number of patients at each facility with larger bubbles corresponding to larger facilities. The horizontal position notes the untrained exit sum score for all questions against the facility, and the vertical position notes the trained exit sum score at the same facility. TEC, trained exit client; UEC, untrained exit client.

constantly reviewing priorities and challenges for optimal engagement in care.[34 35] Patient experience is a key indicator of healthcare quality for meeting the 95-95-95 targets: delivering services patients need, can access and address wider determinants of poor health. Clinicians and health systems must address needs of patients with HIV from diagnosis to death to ensure healthy ageing and viral suppression. Other outcomes in Zambia[11 36 37] show that lifelong needs vary by facility, highlighting the importance of metrics that measure patient experience accurately. We have shown that it is feasible to involve patients in assessing the quality of care and this could potentially lead to involvement of patients in the redesign of healthcare services.

Because HIV care is longitudinal, SPs, who are often used to evaluate episodic care, require highly skilled people to pose as a simulated patient making one visit to multiple clinics, posing practical implementation challenges in our setting.[19–24] Contrary to SPs, we evaluated care quality without using simulated patients and administered the survey once among people in long-term care. Using real patients instead of simulated ones drawn from outside the true patient population, we would argue, made our TEC approach more applicable and reproducible in clinical settings. We were able to record HCW behaviour in a typical HIV context using this concealment method, potentially reducing the impact of the Hawthorne effect. Our TECs also consistently identified more lapses in care, potentially reducing social desirability bias and ability to identify issues at the facility. Even though training takes

time, the increased quality of our measurement allows one to perform fewer surveys. With traditional approaches like exit surveys, one would require a larger sample size, but this does not address bias.[38]

Our findings are consistent with a study done in South Africa which found that non-clinical dimensions of care play a bigger role in determining an overall satisfactory experience for SPs when compared with untrained patients.[38] However, our findings may contradict previous suggestions that tailoring support to individuals to build skills and confidence through patient activation can lead to trained/informed patients reporting a better experience than untrained/uninformed.[39] TECs cared about the following non-clinical aspects of care: rude providers, being satisfied with HIV care providers and spending enough time with providers. This finding is consistent with a previous study in Zambia, where patients reported rude HCWs deterring HIV care engagement.[8 10 11] This could mean that studies assessing patient experience with TEC could focus on a few questions to save time and resources. Questions like 'Did you pick up your medicine or laboratory results at your visit?' may not add much to a TEC survey because they are definitive, and training appears to influence subjective care dimensions.

Female TECs were generally more critical about the care they received and would likely provide a more accurate reflection of the health system, possibly because they have better health-seeking behaviour than men, which may be strongly influenced by local gender norms and health service structures designed to engage women of reproductive age.[40] There is some consistency with other findings that women may be more interested in their care than men, especially in facilities that provide integrated services for women and their children.[9 41] Despite longer wait times, women were more satisfied with integrated facilities.[42] In addition, middle-aged people between 40 and 50 years benefited the most from training. Compared with older people over 50 years, younger people under 30 years were less satisfied with the care they received and often felt they were not greeted by a HCW during their visit. This finding is consistent with cultural norms where younger people are less respected.[43] Given the current strategy of targeting young people, who account for most new infections [1 34], these findings suggest an important new approach to identifying what young people value most. Education level was among the strongest predictors of patient experience feedback. Well-educated patients were found to have a less critical/better HIV care visit experience compared with participants with lower levels of educational attainment. This difference in care experience report may be associated, at least in part, with the HCW perception of the patient in the facility. Research conducted in Nigeria discovered that people with higher levels of education are frequently given better and more considerate treatment by HCWs, hence limited by a form of discrimination/socioeconomic status bias.[44 45]

The observed effect of training on patient experience is likely multifaceted potentially stemming from increased

attention and recall to the exit survey items which solicited a feeling of empowerment to be more critical of the care received. In future studies, patient activation should be measured as an outcome to see how training changes the patient's engagement with their care over time.[39] Further research is required into why female TECs reported poorer experiences with care than men. Other studies that have used SPs to assess medical students' performance showed that women were more critical on certain aspects of care. These studies also recommend matching of SPs to clinicians by sex,[46] something we were not able to do given the nature of our study in primary health facilities where we assessed interpersonal communication with HCWs at all levels. Perhaps our findings call for more investigation into the integration of women's services, such as family planning and children's services with HIV care given some studies have shown this can improve patient satisfaction.[42]

### Limitations

Our findings should be interpreted with caution due to the following limitations. Because this was the first time such a study was done, we recruited educated participants who were able to read and write, perceived to have good recall ability and were able to comprehend things. Our study was only done in Lusaka province in facilities that were largely urban except for one facility which was periurban; hence, it is hard to generalise these findings. Another limitation in our approach is the one-time cross-sectional nature of our measurements among people in long-term HIV care. If more measures were collected from each TEC, we may well see them being activated in a way that results in an improvement in their experience based on the skills they develop to seek better care from providers which ultimately would improve their retention in care. Despite its limitations, the TEC method provides valuable information about healthcare quality, even though it is limited to situations where 'walk-ins' are permitted. Our approach only focused on real patients accessing care and we did not manipulate any patient files, so it is possible that some TECs were known to the facility as patients accessing chronic care. Our approach does require a trained interviewer to speak with TECs after their visits, but this is not any different to what already exists. In future, it may be worth using the domains in the national HIV guidelines as the gold standard, but we did not do this as our aim was to come up with a low-cost approach that can easily be rolled out. In addition, the concept of patient-centred care is still catching on in Zambia. Our TEC approach can be used to further the knowledge in provider attitudes to other relatively new approaches to delivering quality HIV care such as differentiated service delivery for stable patients by assessing whether HCWs follow guidelines when offering this.[35] We also see an opportunity to assess provider–patient communication of viral load laboratory results by use of a universal script for each TEC to assess if they are communicated to and if unsuppressed but adherent, what procedures followed.

### CONCLUSION

TEC offers pragmatic methods for health systems in low-income countries to assess non-clinical dimensions of care (communication, respect and autonomy) which are grounded on the concept of health system responsiveness and could be critical to the transformation of low-quality health systems to high-quality ones.[47] Hawthorne effects and social desirability biases may be mitigated using TECs. We were able to capture HCWs' behaviour in a normal day-to-day low/middle-income setting using similar approaches recommended by King and colleagues that minimise harm to HCWs and SPs.[15] Our findings suggest that TECs provide a more critical appraisal of some aspects of the quality of HIV care. It provides new insights in the Zambian context on what patients value when they interact with the health system. This could be important given the need to reduce LTFU among new ART clients who disengage within the first 6 months of treatment[48] due to a bad first encounter with the health system. Our TEC approach could be used to assess re-engagement interventions. The fact that TECs had a better understanding of the items solicited or felt empowered to be more critical shows that the training we provided worked. This low-cost method could be reproduced in other routine settings and presents an opportunity to further institutionalise patient-centred care by evaluating what happens at the point of contact between the patient, the health facility and the health provider. The implications are that it provides an opportunity to improve HIV care, meet patients' expectations and can serve as a monitoring tool for healthcare performance. Coupled with the recent approaches to client-led monitoring in HIV care, our approach can be used to enhance decision-making that considers patients' involvement.

**Author affiliations**
[1]Implementation Science Unit, Center for Infectious Disease Research in Zambia, Lusaka, Zambia
[2]Department of Public Health, Environments and Society, London School of Hygiene and Tropical Medicine Faculty of Public Health and Policy, London, UK
[3]Internal Medicine, Washington University in St Louis School of Medicine, St Louis, Missouri, USA
[4]Department of International Health, Johns Hopkins University Bloomberg School of Public Health, Baltimore, Maryland, USA
[5]Social and Behavioural Science Research Group, Center for Infectious Disease Research in Zambia, Lusaka, Zambia
[6]Faculty of Epidemiology and Population Health, London School of Hygiene and Tropical Medicine, London, UK
[7]Department of Medicine, The University of Alabama at Birmingham, Birmingham, Alabama, USA
[8]Center for Innovation in Global Health, Georgetown University Medical Center, Washington, District of Columbia, USA

**Acknowledgements** We would like to thank recipients of care for taking part in this study. We would like to thank the Zambian Ministry of Health, the US President's Emergency Plan for AIDS Relief (PEPFAR) through the US Centers for Disease Control and Prevention/Zambia (CDC) and the Centre for Infectious Disease Research in Zambia for their leadership in ensuring those in HIV care continue to receive life-saving HIV treatment. We would also like to thank the healthcare workers who continued to faithfully deliver HIV care in the face of the COVID-19 pandemic.

**Contributors** KS is the guarantor and lead author, conducted all analyses, led data management activities and field coordination of activities, and designed data collection tools. JMP was responsible for field coordination of data, assisted with analysis and revising it critically for important intellectual content, and designed data collection tools. AM assisted with analysis, framing and revising it critically for important intellectual content. BR gave final approval for publication, assisted with framing and revising it critically for important intellectual content. CB was responsible for field coordination of data quality processes and data acquisition. IE-W drafted statistical analysis plan, and assisted with conceptualisation and interpretation of data. JM assisted with data acquisition and cleaning, and field coordination of data quality processes. SS led intervention implementation, project administration and data curation. LKB, NM and AS cognitively tested data collection tools, assisted with conceptualisation and underlying data processes, and assisted with writing and data interpretation. DM and MM-M advised on implementation details. AW and JH assisted with framing and revising it critically for important intellectual content. CB-M was the lead for underlying data processes and assisted with funding acquisition. CH did funding acquisition, assisted with conceptualisation and advised regarding intervention implementation details. ITS did funding acquisition, and assisted with conceptualisation and manuscript writing. EG did funding acquisition, led conceptualisation, advised on all analyses and made final approval for publication.

**Funding** This work was supported, in whole or in part, by the Bill & Melinda Gates Foundation (INV-010563). Under the grant conditions of the Foundation, a Creative Commons Attribution 4.0 Generic License has already been assigned to the Author Accepted Manuscript version that might arise from this submission. This work was also supported by the National Center for Advancing Translational Sciences (grant KL2 TR002346 to AM) and the National Institute of Allergy and Infectious Diseases (grant K24 AI134413 to EG).

**Disclaimer** The contents of this paper are the sole responsibility of the authors and do not necessarily reflect the views of the Bill & Melinda Gates Foundation. The funders had no role in study design, data collection and analysis, decision to publish or preparation of the manuscript.

**Competing interests** None declared.

**Patient and public involvement** Patients and/or the public were involved in the design, or conduct, or reporting, or dissemination plans of this research. Refer to the Methods section for further details.

**Patient consent for publication** Not required.

**Ethics approval** This study involved human participants and ethics approval to conduct this research was granted by the Zambian Ministry of Health, National Health Research Authority, and the institutional review boards of the University of Zambia (008-03-19), the University of Alabama at Birmingham (300003282) and the London School of Hygiene and Tropical Medicine (21384). Participants gave informed consent to participate in the study before taking part.

**Provenance and peer review** Not commissioned; externally peer reviewed.

**Data availability statement** Data are available upon reasonable request. The Government of Zambia allows data sharing when applicable local conditions are satisfied. In this case, the data from the study will be made available to any interested researchers upon request. The CIDRZ Ethics and Compliance Committee is responsible for approving such requests. To request data access, one must write to the Secretary to the Committee/Head of Research Operations, Mrs Hope Chinganya (Hope.Chinganya@cidrz.org), mentioning the intended use for the data. The committee will then facilitate review and authorisation to release the data as requested. Data requests must include contact information, a research project title and a description of the intended use.

**ORCID iDs**
Kombatende Sikombe http://orcid.org/0000-0002-8187-8661
Jake M Pry http://orcid.org/0000-0001-6312-4420
Aaloke Mody http://orcid.org/0000-0003-3787-365X
Ingrid Eshun-Wilson http://orcid.org/0000-0002-4049-9868

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
