## [Reviewer comments · BMJ Open]

ARTICLE DETAILS

TITLE (PROVISIONAL)	A Comparison of Patient Exit Interviews to Unannounced Standardised Patients for Assessing HIV Service Delivery in Zambia nested within a Cluster Randomised Trial
AUTHORS	Sikombe, Kombatende; Pry, Jake; Mody, Aaloke; Rice, Brian; Bukankala, Chama; Eshun-Wilson, Ingrid; Mutale, Jacob; Simbeza, Sandra; Beres, Laura K; Mukamba, Njekwa; Mukumbwa-Mwenechanya, Mpande; Mwamba, Daniel; Sharma, Anjali; Wringe, Alison; Hargreaves, James; Bolton-Moore, Carolyn; Holmes, Charles; Sikazwe, Izukanji; Geng, Elvin

VERSION 1 – REVIEW

REVIEWER	MacDonald Hompashe, Dumisani University of Fort Hare, Economics
REVIEW RETURNED	21-Nov-2022

GENERAL COMMENTS	This is a very innovative paper and easy to read. I am impressed by the trained exit clients approach instead of simulated patients. However, the authors did not clarify how the training was conducted. Were the patients trained as they came to the facilities in different groups and in other days, weeks, or months? I would like the authors to address this in the revised version.
--

REVIEWER	Munyayi, Farai University of the Western Cape
REVIEW RETURNED	21-Jan-2023

GENERAL COMMENTS	The objective of the study is not consistent in the abstract vs methods section, the abstract states that the objective is to evaluate sub-optimal patient experiences in HIV care, whilst in the methods it states that the study seeks to compare two different methods for assessing patient experience. The authors may consider clarifying which specific question the study seeks to answer. The abstract should also clearly state the study design/ methods used, whilst in the abstract it states Setting as "Cross-sectional surveys in 16..." in the Methods section it states that this study was within a stepped wedge cluster randomised trial, intervention/experiemtal study? Consider clarifying the "systematically sampled" term in the abstract, what was the sampling method used? Whilst the authors mention that the participants only received a snack during survey administration and no other financial incentive, later in the paper they also mention that TEC participants were given K100 (approx. \$5). Were the sub-groups for the analysis pre-defined or they were considered after the results were available, at analysis?
---

	Whilst the 95-95-95 targets are addressed in the discussion section, I think having them as part of the introduction/background would be great in highlighting their importance, and then discussing the related meaning with the findings from the study in the discussion section. Please also clarify the use of Ref 45 in line 444.
--	---

VERSION 1 – AUTHOR RESPONSE

Response to Reviewers:

3) This is a very innovative paper and easy to read. I am impressed by the trained exit clients approach instead of simulated patients. However, the authors did not clarify how the training was conducted. Were the patients trained as they came to the facilities in different groups and in other days, weeks, or months? I would like the authors to address this in the revised version.

We thank the reviewers for their feedback and interest in this new approach. We have addressed the concerns around a better description of how training was conducted. We describe the details of who conducted the training from line 213-217. We provide more details on when recruitment and training of trained exit clients took place in line 240-247. We have also included in line 231 that the training sessions were one-on-one sessions conducted on the day of their visit (line 230). Further details on when the TEC completed the survey in a private area after training and their visit are provided in line 237 and 239.

4) The objective of the study is not consistent in the abstract vs methods section, the abstract states that the objective is to evaluate sub-optimal patient experiences in HIV care, whilst in the methods it states that the study seeks to compare two different methods for assessing patient experience. The authors may consider clarifying which specific question the study seeks to answer.

Thank you for the opportunity to make our objectives clearer. We have revised the objectives in the abstract to include “To evaluate the use of an adapted standardised patient approach (e.g., mystery clients) to measure sub-optimal patient experiences in HIV care (e.g., unfriendly interactions with health care workers [HCW], long-waiting times, and lost laboratory results).” This is captured in line 48.

5) The abstract should also clearly state the study design/ methods used, whilst in the abstract it states Setting as "Cross-sectional surveys in 16...." in the Methods section it states that this study was within a stepped wedge cluster randomised trial, intervention/experimental study?

Thank you for the opportunity to clarify. This manuscript reports on a sub-study that is nested within a larger stepped-wedge cluster randomized trial (which is being reported separately). However, in this sub-study, we used surveys that were collected cross-sectionally. We made revisions to the study objectives to help to make these distinctions more clear. In addition, we have also made a change to our language under the outcome measures of the abstract in line 61 to include the following statement: “We compared patient experience among patients that received a brief training prior to their care visit (explaining each patient experience construct in the exit survey, being anonymous, not altering behaviour or manipulating interactions) with those who did not undergo training on the instrument prior to their visit”. Further clarity has been provided in line 186-188 in the study design and setting under the methods section.

6) Consider clarifying the "systematically sampled" term in the abstract, what was the sampling method used?

We have made changes to line 58 of the abstract to reflect that we used a systematic sample (every kth file) of patients visiting the facility. This change has also been captured in line 196.

7) Whilst the authors mention that the participants only received a snack during survey administration and no other financial incentive, later in the paper they also mention that TEC participants were given K100 (approx. \$5).

We thank the reviewers for the opportunity to clarify. TEC participants were given K100 (~\$5) due to the additional time required for training as well as a light snack during the survey administration. Exit participants did not receive anything. We have rewritten this section from line 246-247 to clarify these distinctions.

8) Were the sub-groups for the analysis pre-defined or they were considered after the results were available, at analysis?

Based on literature on patient experience and HIV, we used pre-defined variables for our sub-group analysis. This has been captured in line 262.

9) Whilst the 95-95-95 targets are addressed in the discussion section, I think having them as part of the introduction/background would be great in highlighting their importance, and then discussing the related meaning with the findings from the study in the discussion section.

Thank you for this suggestion. We have added this in line 139-141 of the introduction. We have linked the importance of more patient centered care to attaining global 95-95-95 targets, especially retention.

10) Please also clarify the use of Ref 45 in line 444.

The reference list has now been updated and we've provided more context to the use of this reference. We found ref 45 by King et al to be a good guide on how to carry out SP assessments in low middle income countries. It describes how SPs should be done and we have captured this useful reference in line 445-447.

VERSION 2 – REVIEW

REVIEWER	Munyayi, Farai University of the Western Cape
REVIEW RETURNED	30-Mar-2023
GENERAL COMMENTS	Thank you for the opportunity to review this revised manuscript. I am satisfied with the responses and revisions done by the authors.